# Leadership for success in transforming medical abortion policy in Canada

**Brigid Dineley**[1]*, **Sarah Munro**[1,2], **Wendy V. Norman**[3,4]

1 Department of Obstetrics & Gynecology, University of British Columbia, Vancouver, Canada, 2 Centre for Health Evaluation and Outcome Sciences (CHÉOS), Vancouver, Canada, 3 Department of Family Practice, University of British Columbia, Vancouver, Canada, 4 Faculty of Public Health & Policy, London School of Hygiene & Tropical Medicine, London, England

* brigid.dineley@cw.bc.ca

**Data Availability Statement:** All relevant data are within the paper.

**Funding:** This study is supported by grants received by Wendy Norman from Canadian Institutes for Health Research, CIHR (PHE148161),

## Abstract

### Objectives

Mifepristone was approved for use in medical abortion by Health Canada in 2015. Approval was accompanied by regulations that prohibited pharmacist dispensing of the medication. Reproductive health advocates in Canada recognized this regulation would limit access to medical abortion and successfully worked to have this regulation removed in 2017. The purpose of this study was to assess the leadership involved in changing these regulations so that the success may be replicated by other groups advocating for health policy change.

### Methods

This study involved a mixed methods instrumental design in the context of British Columbia, Canada. Our data collection included: a) interviews with seven key individuals, representing the organizations that worked in concert for change to Canadian mifepristone regulations, and b) document analysis of press articles, correspondence, briefing notes, and meeting minutes. We conducted a thematic analysis of transcripts of audio-recorded interviews. We identified strengths and weaknesses of the team dynamic using the Develop Coalitions, Achieve Results and Systems Transformation domains of the LEADS Framework.

### Results

Our analysis of participant interviews indicates that autonomy, shared values, and clarity in communication were integral to the success of the group's work. Analysis using the LEADS Framework showed that individuals possessed many of the capabilities identified as being necessary for successful health policy leadership. A lack of post-project assessment was identified as a possible limitation and could be incorporated in future work to strengthen dynamics especially when a desired outcome is not achieved. Document analysis provided a clear time-line of the work completed and suggested that strong communication between team members was another key to success.

in partnership with Michael Smith Foundation for Health Research (Award #16743). In kind and team infrastructure support was provided by the Women's Health Research Institute of British Columbia Women's Hospital and Health Centre of the Provincial Health Services Authority of British Columbia. SM is supported as a Trainee and a Scholar of the Michael Smith Foundation for Health Research [16603, 18270]. WVN is supported as a Scholar of the Michael Smith Foundation for Health Research [2012-5139 (HSR)], and as an Applied Public Health Research Chair by the Canadian Institutes of Health Research [CPP-329455-107837]. The funders had no role in study design, data collection and analysis, decision to publish or preparation of the manuscript.

**Competing interests:** The authors have declared that no competing interests exist.

## Conclusions

The results of our analysis of the interviews and documents provide valuable insight into the workings of a successful group committed to a common goal. The existing collegial and trusting relationships between key stakeholders allowed for interdisciplinary collaboration, rapid mobilization, and identification of issues that facilitated successful Canadian global-first deregulation of mifepristone dispensing.

## Introduction

In Canada, abortion was decriminalized in 1988, and it remains the only country in the world to have fully enacted decriminalization. [1,2] There are currently no Canadian criminal laws restricting abortion access. For example, this includes that Canada has no criminal law stipulating restrictions on gestational age and there are no criminal laws which require authorization from a medical board, or a specified number or type of physician, prior to obtaining a procedure. [2] In Canada, abortion care is regulated as any other medical service; governed by provincial health professional regulators and health system authorities. Following decriminalization there was an attempt to restrict access through provincial channels, but currently the only regulations that exist are designed to improve access to abortion services. [2] Despite this, The UN Human Rights Commissioner's report identified that access to abortion in Canada is known to be unequally distributed. [3] This inequity is caused by the monetary cost of some abortion services, the challenges with awareness of how to access services and the geography of providers. [2] The majority of providers are located in urban centres and there is minimal provision in rural and remote locations. [4] Due to this clustering of providers, individuals seeking abortion care who do not reside in these urban centres must travel to access care, resulting in increased time away from work, family obligations, social supports, as well as substantial monetary costs for travel and accommodation. [4–6]

Medical abortion offers a potential solution to the geography of providers, with the ability to provide abortions without the need for a surgical centre or surgical training. [7] Medical abortion involves two medications: mifepristone and misoprostol, which are taken 24–48 hours apart. It is a safe and effective way to terminate a pregnancy up to 70 days gestational age. [8] The medications are dispensed together in one box, in separate colour coded packages, and are distributed in Canada under the brand name Mifegymiso® by Celopharma. [8] This medication could address geography-related access problems by being distributed through primary care networks as well as through telemedicine services.

Mifepristone was first approved for use in Canada in 2015. When Health Canada approved the medication several restrictions to dispensing were put in place. [7,8] Most notably, physicians were required to dispense the medication directly to patients and pharmacists were restricted from this role. [9] Other restrictions included observed ingestion and mandatory completion of education modules for all providers. [7] These restrictions were similar to those introduced in other countries such as the US, where supervised ingestion was mandated. [10] Australia also continues to experience a variety of regulatory barriers depending on geographic location and research has indicated that these regulations limit provision of medical abortion by primary care professionals. [11] When the Canadian restrictions were announced, advocates for abortion access raised concerns that they would limit particularly the potential for primary care provision and thus access to abortion. Further, there was no evidence that the restrictions would improve patient safety. [7] Provincial frameworks also discouraged or

restricted physician dispensing. In BC, for instance, physician dispensing would require access to Pharmanet (a central data system tracking every prescription in the province) and special permission from the College of Physicians and Surgeons of British Columbia (CPSBC). [12,13] Physician-only dispensing of mifepristone was perceived by advocates to be an unnecessary barrier that, if removed, would dramatically impact patient access to abortion by encouraging uptake of the provision of medical abortion care among prescribers and pharmacists.

The purpose of this study was to investigate the leadership skills that were used to advocate for changing, and successfully removing, the restrictions on pharmacist dispensing of mifepristone in BC over the course of twelve months in 2016–2017. This work was done by physicians, pharmacists, public health experts and hospital executives who worked in parallel, engaging with each other as needed while each working within their own organizations and areas of expertise to improve access to necessary health care for individuals in BC. This work will be termed "the project" for the remainder of this paper. Our investigation of these leadership skills is important to provide insight into the functioning of a successful team and a potential guide to skill development for other teams working on policy.

## Methods

This study employed a single site, mixed methods instrumental design to assess the leadership involved in the project using the Lead self, Engage others, Achieve results, Develop Coalitions and Systems Transformation (LEADS) framework.

### Analysis framework

The LEADS framework provides a summary of the key skills, attitudes, and qualities necessary to foster change while working within the Canadian health system. [14] The LEADS Collaborative is a partnership between the Canadian College of Health Leaders the Canadian Health Leadership Network, Royal Roads University and Dr. Graham Dickson. [14] The framework was developed in BC in conjunction with several large health sector employers and the Health Care Leaders Association of BC. The framework was initially designed specifically for BC and was then scaled up to incorporate a Canadian wide approach. [15] The LEADS framework consists of 5 domains: Lead Self, Engage Others, Achieve Results, Develop Coalitions, and Systems Transformation. [14] The framework assesses leadership from an individual to systems level.

We chose the LEADS framework to analyze our data as it directly applies to the Canadian setting in which the project was undertaken. We felt that it would provide a relevant evaluation of leadership qualities given that it was designed by Canadian health care institutions for use in the Canadian health care system, to document the leadership attributes necessary to drive change within the Canadian health care system. The framework stresses that leadership and management skills must combine together in order to stimulate and sustain change. Use of the framework has been shown to foster stakeholder engagement in systems change by employing a common language and set of standards. [15] Leadership is necessary to foster an environment where positive, evidence-based change can be introduced by all levels of a team.[16] Our focus was on leadership skills used by the individuals in the team that facilitated a successful policy change and therefore, because we were not explicitly exploring the policy change itself, we did not feel it would be appropriate to use a policy framework for analysis.

We felt that the Leads Self domain was not as applicable to the study as it emphasizes skills of self-awareness and emotional intelligence that we felt would be difficult to assess through an interview process. We also identified that the Engage Others domain was more applicable to formal team structures, which was not the way the individuals involved in the project were

organized. Through research team discussion and deliberation, we determined the most relevant domains to be: Achieve Results, Develop Coalitions, and Systems Transformation.

## Recruitment

BD identified WN as a key stakeholder and proposed the study. WN's role as the Director of the Contraception & Abortion Research Team, CART-GRAC, provided a unique opportunity to assist with recruitment, design and analysis. CART-GRAC is a Canadian multi-disciplinary team that carries out research on topics identified by policy leaders, community groups, and health care providers with the objective of improving access to family planning care for all Canadians. Their work on mifepristone implementation in Canada is supported by a grant from CIHR (PHE148161), in partnership with the Michael Smith Foundation for Health Research (Award #16743).

BD interviewed key stakeholders involved in the project in BC, identified through snowball sampling. We (BD, WN) initially developed a list of potential participants and expanded it as input from these key stakeholders was incorporated. We contacted potential participants via email and asked if they would be interested in completing an interview to discuss their role in the project. In the email we gave a brief description of the goal of the study and explained that we would be employing the LEADS Framework to perform our analysis. We obtained written consent at the time of the interviews. Inclusion criteria included any individuals identified by another member of the group to have had a significant role in the project. We reviewed the list with each participant at the time of their interview to solicit feedback and to determine if any individuals had been missed. Our participant list was determined to be complete when no further names were generated by the current participants. Study participants were assigned a random, unique three digit study identifier.

## Interviews

Written consent was obtained prior to all interviews which were recorded by the author and transcribed. We completed interviews in person, by email, and over the phone. The lead author (BD) carried out all interviews. The interview questions were based on a previously employed interview guide used by the CART team to assess stakeholder engagement in mifepristone implementation work and modified to directly assess the domains of the LEADS framework. [17]Interview questions were developed with a priori knowledge of the team members and their multi-disciplinary nature. We employed a semi-structured interview guide for all participants, ensuring that we addressed all domains with all participants. The interview guide also included a description of the LEADS framework. Example questions included:

1. Describe your role in the work that was done to remove the requirement that physicians dispense mifepristone:

    a. When and how did you get involved?

    b. Who was on the team, was there a clearly defined leader?

    c. What was your experience working with this team?

    d. Did you have any role in engaging hospital executive members

    e. How was progress on the project measured?

2. Describe the process of working with multiple stakeholders to achieve results

    a. How were these stakeholders identified?

b. How important was inter-disciplinary collaboration to the success of this project?

c. Was communication uni or bi-directional?

d. Was the exchange of ideas between groups easy or difficult?

### Analysis

BD analyzed the interviews using the LEADS framework, specifically looking at the domains of Achieve Results, Develop Coalitions, and Systems Transformation. Each interview was assessed through the lens of each domain. We documented the presence and absence of the key capabilities of each domain in each interview. Using this method, we were able to identify which capabilities were present and which were absent. Strengths and weaknesses of the team dynamic were identified. Extensive field notes were taken during and after each interview to assist in analysis.

In addition to interviews with key stakeholders, textual analysis of a variety of documents was conducted to create a cohesive timeline of events and to assess elements of teamwork. These documents included: press articles, submissions to Health Canada, emails between team members, briefing notes for the College of Physicians and Surgeons of BC board meetings, and Proceedings from the annual CART research meeting at BC Women's Hospital. [9,18–25]

Ethics approval was obtained through the University of British Columbia Children's and Women's Hospital ethics review board (H16-01006). All documents reviewed were released to the author in accordance with the Freedom of Information act.

As WN was both a participant in this study and a member of the research team. Bias was mitigated in study design by development of the interview questions (SM, BD) independent of input from WN. As well, factual information provided in her interview (such as the timeline of the project) was verified via document analysis of information that was obtained from other participants as well as through the interviews with other team members. In order to ensure that there was no bias in the identification of stakeholders, the other participants were asked to suggest additional potential individuals who played a meaningful role in the project. Finally, analysis of the data was carried out by BD and SM without input by WN.

## Results

### Identification of stakeholders

In total we completed seven interviews. See Fig 1 for an illustration of the recruitment process.

### Document analysis

Fig 2 outlines the timeline of policy change.

### Leadership analysis

The LEADS domains were used to analyse the transcripts. Fig 3 explains the domains and their respective capabilities.

**Develop coalitions.** According to the LEADS Framework, collaboration is a central part of making changes to health policy.[14] Develop Coalitions is a key domain in successful leadership and involves an understanding of the Theory of Collaborative Advantage, which stresses that collaborations must be actively built and sustained in order to be successful. [26] The four capabilities of the Develop Coalitions domain are: a) purposefully building partnerships and networks to create results, b) mobilizing knowledge, c) demonstrating a commitment to customers and service, and c) successfully navigating socio-political environments. [26]

## Initial Interview

## Second Interviews

## Third Interviews

**WN**

Physician Regulation

Pharmacy Regulation

Pharmacy Research

Hospital Administration

Government

Pharmacy Education

**Fig 1. Recruitment timeline.**

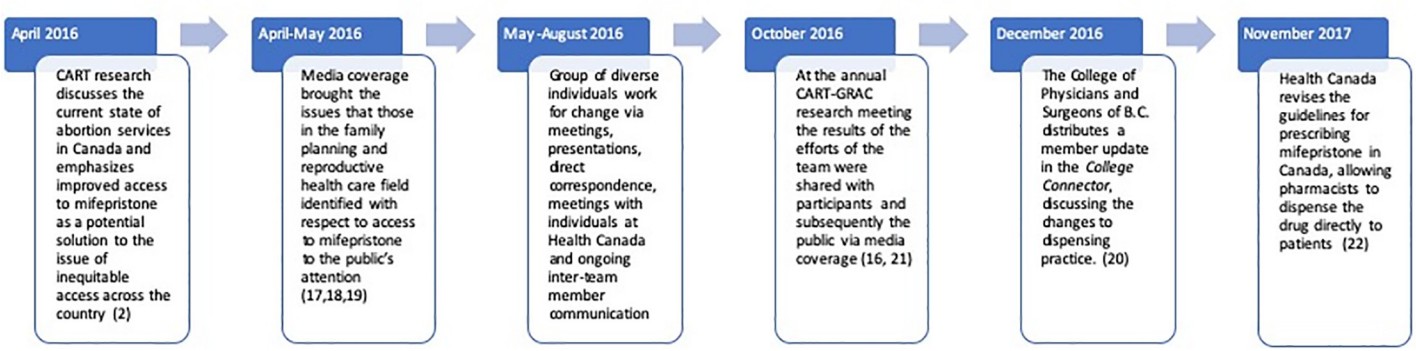

| April 2016 | April-May 2016 | May-August 2016 | October 2016 | December 2016 | November 2017 |
|---|---|---|---|---|---|
| CART research discusses the current state of abortion services in Canada and emphasizes improved access to mifepristone as a potential solution to the issue of inequitable access across the country (2) | Media coverage brought the issues that those in the family planning and reproductive health care field identified with respect to access to mifepristone to the public's attention (17,18,19) | Group of diverse individuals work for change via meetings, presentations, direct correspondence, meetings with individuals at Health Canada and ongoing inter-team member communication | At the annual CART-GRAC research meeting the results of the efforts of the team were shared with participants and subsequently the public via media coverage (16, 21) | The College of Physicians and Surgeons of B.C. distributes a member update in the *College Connector*, discussing the changes to dispensing practice. (20) | Health Canada revises the guidelines for prescribing mifepristone in Canada, allowing pharmacists to dispense the drug directly to patients (22) |

**Fig 2. Timeline of policy change.**

**Develop Coalitions**

1. Purposefully building partnerships and networks to created results
2. Mobilizing knowledge
3. Demonstrating a commitment to customers
4. Navigating socio-political environments

**Achieve Results**

1. Setting Direction
2. Strategically aligning decisions with vision, values and evidence
3. Taking action to implement decisions
4. Assessing and evaluating

**Systems Transformation**

1. Demonstrating systems/critical thinking
2. Encouraging and supporting innovation
3. Orienting strategically to the future
4. Orchestrating change

**Fig 3. LEADS domains and respective capabilities analyzed[12-14].**

There were multiple interlocking coalitions created during the course of the project. Our analysis of the interviews revealed collaboration between the CART team, the College of Physicians and Surgeons of BC, the College of Pharmacists of BC, the executive at BC Women's Hospital, and members of provincial and federal governments. These overlapping coalitions will be evaluated on the basis of each of the four elements of the Develop Coalitions domains.

**Develop coalitions: Purposefully building partnerships.** Our analysis of interviews suggest that the type of collaboration pursued during the project fit into the category of joint ventures; where the members of the groups were working together towards the common goal of removing the restrictions on the dispensing of mifepristone. These collaborations were highly strategic. The CART team identified the key stakeholders that would help to move the project forward. They were then able to provide the best medical evidence to support the regulatory bodies in communicating with Health Canada to produce change.

"Our research team was able to support them (regulatory bodies) taking a stand against the federal regulator founded by the best evidence as well as on their regulatory structure." (P742)

Importantly, multiple individuals we interviewed communicated that these partnerships had been formed in previous projects and so trust and mutual respect were present, which made collaboration and partnership building faster and simpler.

"For me it was like dropping the stone in the puddle because we have these established connections already . . . it was pretty easy to get a message out." (P328)

"There is trust and credibility . . . comfort that someone isn't trying to get something done that doesn't align with our values." (P390)

Partnerships used in the project were selected with clear intent; every member had resources in the form of knowledge, expertise, advocacy, government contacts, and/or regulatory authority.

**Develop coalitions: Mobilizing knowledge.** Our analysis of the interviews also demonstrated themes of knowledge transfer and evidence sharing. Specifically, the CART team was cited as being able to provide medical evidence to the board of the College of Physicians and Surgeons of BC as well as to the board of the College of Pharmacists of BC on mifepristone, medical abortion, and the lack of access to abortion care for many Canadians. Additionally, knowledge of the policy of medication prescribing practices was communicated by the College of Pharmacists of BC to the other coalition members. These key education points were combined to achieve both "Know-how," or how the policy should be changed, and "Know-why," or what evidence supported the removal of the barriers to pharmacists dispensing mifepristone. It was through the purposeful partnerships that were formed that the right knowledge and evidence could be shared and then mobilized to produce change within Health Canada. Specifically, this was communicated in interviews when P742 and P413 described giving presentations to the regulatory bodies that shared the most recent evidence on the safety of pharmacist dispensing of mifepristone.

Additionally, research findings, or knowledge, were mobilized by the CART team and communicated to the media. The CART team was able to use knowledge translation to take their research data that indicated that the restrictions in place were limiting provision of medical abortion and bring it to a new set of knowledge users. Through a series of news articles [20–22,24] the team was able to engage the public and the wider health care community with the evidence supporting a proposed repeal of Health Canada's regulations.

**Develop coalitions: Demonstrating a commitment to customers and service.** Participants' attitudes and experiences demonstrated a commitment to the delivery of equitable reproductive care to individuals in BC. The impetus for the project was the data found in CART's initial research article that demonstrated a lack of access to abortion care for many individuals living outside of urban communities. [4] The hospital executive at BC Women's Hospital mobilized their value of reproductive choice for all women to support the project team and provide support and testimony of impact to further highlight the critical nature of the restriction on mifepristone access. It was clear from the interviews that a dedication to improving access for individuals seeking abortion care was a top priority for the project: "It's about good care" (P390).

**Develop coalitions: Navigating socio-political environments.** Another key theme expressed by participants during the interviews was that of autonomy. We found that participants identified that team members were encouraged to work towards the clear goal of the project but did not feel they were being closely monitored or supervised. Communication between stakeholders occurred on a regular basis and was bi-directional.

"Communication processes were set up so that we are always informed of where things were at." (P519)

This high level of autonomy could be attributed to the pre-existing relationships between the members of the team and the trust and mutual respect that was already present. One participant summarized this with the following quotation:

"Collegial, professional, trusting relationships were already in place, enabling the rapid exchange and initiation of ideas and recommendations for change." (P629)

**Achieve results.**   In the LEADS Framework, achieving results is a key component of being a successful leader. [14] Specifically, the framework emphasizes that strong leaders may act before all individuals in an organization are on board and are able to mobilize the resources available to them to make public health change. [27] The four key capabilities of the Achieve Results domain are: a) setting direction, b) strategically aligning decisions with vision, values, and evidence, b) taking action to implement decisions, and c) assessing and evaluating. [27]

Our analysis of interviews and documents suggests that the project was successful demonstrating all four of these capabilities, and in achieving three main results: providing evidence to support pharmacist dispensing of mifepristone, providing physicians and pharmacists in BC with regulatory support to work outside of Health Canada regulations, and contributing to changes to the Health Canada regulations for mifepristone.

**Achieve results: Setting direction.**   Participants suggested an experience of shared and aligned values. Specifically, the values of the many organizations involved lined up to make the project successful. The direction of the organizations supported pharmacists, physicians, and individuals in BC seeking abortion care. The leadership at BC Women's Hospital had already worked diligently to foster an environment that championed reproductive choice throughout an individual's reproductive lifetime. The College of Pharmacists of BC had a clear vision that it was the role of the provincial regulator, rather than that of Health Canada to regulate pharmacists' dispensing practice. The College of Pharmacists of BC also expressed their dedication to equitable access to all medications for the individuals in BC. The College of Physicians and Surgeons of BC were concerned about access to abortion care and the undue burdens placed on both individuals seeking care and the physicians providing that care. The UBC Faculty of Pharmaceutical Science also demonstrated values of justice and equity, among others. Finally, CART's goal also aligned as they aimed to use evidence to support health policy and service decisions able to provide Canadians the ability to plan their pregnancies. A key to success of the project was the alignment of the direction and vision of multiple organizations, which ensured the commitment of the parties involved.

**Achieve results: Strategically aligning decisions with vision, values, and evidence.**   This domain stresses the importance of understanding the complex nature of the health care system and "aligning strategy with structure, culture and skills." [27] The interviews demonstrated how the success of the project was dependent on the ability of the leaders involved to create a network working for change made up of many different organizations. In many of the interviews, individuals commented on the clear communication that occurred between the groups: communication was "bi-directional with real-time sharing of information" (P413). The project was able to use the strengths and expertise of each organization, coming together with "The right people . . . the right institution . . . the right political party" (P629) to create change.

**Achieve results: Taking action to implement decisions.**   Through the interviews we were able to gain more information on the actions taken to create change as outlined above in Results. As noted, participants also commented on the bi-directional communication that occurred between the members of the project, with sharing of progress and setbacks among team members occurring on a regular basis. Additionally, decisive action was taken on the

part of the members involved, demonstrated by an immediate response to the Health Canada regulations on mifepristone.

Finally, clarity was stressed as a component that was integral to the success of the team. participants described clarity around evidence, policy and practice that allowed them to focus on results.

> "Clarity was most important, having people in BC who were really able to articulate why change needed to happen, how they could help make that happen and why the barriers that have been conceived . . . were actually doing more harm to patients." (P629)

Within the LEADS Framework, creativity in response to challenges is discussed as a building block of this capability. One participant described that: "knowing who to use and how to use them is creative" (P519), showing that clarity and creativity were both used to achieve change. This participant was referring to the ability to recognize team members' strengths and areas of expertise and make the best use of them as being creative and necessary to success, instead of confining everyone only to their formal roles within an organization. Additionally, another participant described the following:

> "Adaptability and creative ways of interacting with stakeholders and health policy makers in multiple ways and in varied settings (e.g. one-on-one meetings; Board meetings, Conferences etc.) was essential [to success]." (P413)

**Achieve results: Assessing and evaluating.**   This domain was not as relevant to the leadership of the project. The project was self-limited in that it concluded when the desired change to regulation was instated. Ongoing formal performance assessments and metrics were not commented on in the interviews, instead progress was measured by the changes made to the problematic dispensing regulations.

**Systems transformation.**   Finally, the LEADS Framework also states that systems transformation is needed to improve the Canadian health sector. [28] Strong leaders require a deep understanding of the nuanced economic, budgetary, technological, and inter-disciplinary considerations that must align in order to successfully create change in a complex and constantly evolving system. [28] The four key capabilities of the Systems Transformation Domain are: a) demonstrating systems/critical thinking, b) encouraging and supporting innovation, c) orienting strategically to the future and, d) orchestrating change. [28]

**Systems transformation: Demonstrating systems/critical thinking.**   This capability focuses on the understanding of the health care system in which the work is being done (in this case, both BC and Canada) and then using critical thinking skills to determine which health care modalities will work in future health systems. [28] Each participant had in-depth knowledge and experience with their particular domain and then a greater understanding of the way in which this domain fit into the larger provincial and national health system.

> "(We) had a pretty good understanding of how the system is supposed to work . . . the regulation of the professions and who does what is entirely within the domain of the provinces." (P390)

> "It (the regulation) struck us as simply not how medicine and pharmacy work as practiced in BC." (P328)

Participants communicated that this knowledge was one of the reasons that the issues with prohibiting pharmacists from dispensing mifepristone were identified (cost, safety, skill) and

also how the solution to this issue was identified (health professional practices are governed by their provincial "College," not by the federal drug regulator Health Canada).

**Systems transformation: Encouraging and supporting innovation.** The second capability in the systems transformation domain describes the importance of Quality Improvement and using models such as the Plan-Do-Study-Act cycle (PDSA) of innovation. [28,29] The PDSA cycle is a framework created by the National Health System (NHS) in the UK to guide implementation of novel changes in a regulated and controlled fashion. The system is important in leadership because it allows for innovation to be both supported and introduced in a way that encourages ongoing evaluation. [28,29] Participants in this project generally did not coalesce to craft a formalized PDSA cycle for the project. However, some elements were present in the work that was carried out. The team planned an intervention and then carried it out. Similarly, starting from the 2015 announcement of the restrictive mifepristone regulation in Canada, CART planned a national study very similar to a PDSA cycle, the "CART-Mifepristone Implementation Study" [17] (funded in July 2016 as noted above by CIHR and MSFHR). Throughout the study iterative cycles of engagement such as described here with research data being generated and then shared with stakeholders and knowledge users throughout Canada, have been undertaken to present evidence, change policy, and to assess the outcomes. This has been undertaken through news articles [20–22]research papers [17]and through annual meetings of the CART team. [18] One participant also raised the importance of recognition of accomplishments "We really do need to give credit" (P519) when discussing teamwork.

**Systems transformation: Orienting strategically to the future.** An orientation to the future was clearly expressed by many of the participants. Specifically, the CART team recognized that the future of abortion care in Canada had the potential to change significantly with the introduction of mifepristone, but that the full potential of medical abortion would only be realized if change to dispensing regulations were made. The values of the regulatory colleges as well as BC Women's Hospital reflected the future of health care: a system that is patient centred, multi-disciplinary, and collaborative. [30]

**Systems transformation: Championing and orchestrating change.** The final capability of the Systems Transformation domain discusses how leaders are aware and understanding of the relationships between stakeholders and other knowledge users and individuals within the health system. A leader can support and stimulate engagement from a variety of different groups in order to generate change. [28] As described by the participants, multiple organization were involved in the project. Stakeholder and professional engagement were successfully achieved. In the interviews, what became clear was that the members of the project understood that change to medication policy was not a matter that should be addressed only by researchers, physicians, hospital administration, government, or pharmacists independently, but rather a process that can only be successful when all of the above entities are given a seat at the table. When discussing the beginning of the project, participant 1 stated they had:

> "A plan to gather more collaboration and information and connection to understand what the timeline and processes might be moving forward." (P742)

One participant expressed that the engagement of such a wide variety of stakeholders "provided the government with reassurance that it is not just one group with a vested interest" (P519) who is pursuing change and thus the presence of the wide-based coalition helped to provide legitimacy to the project.

## Discussion

The purpose of this study was to investigate the leadership skills that were used to advocate for changing, and successfully removing, the restrictions on pharmacists dispensing mifepristone in BC within the course of twelve months in 2016–2017. We completed document analysis to determine a clear timeline of events and conducted interviews with individuals identified through snowball sampling. We analyzed the interviews using the LEADs framework in an effort to determine what factors contributed to successful policy change. The policy change work was carried out by a group of individuals from a variety of backgrounds including medicine, research, pharmacy, hospital administration, and government.

Our case study identified the characteristics of the stakeholder team that typified the LEADS capabilities of Develop Coalitions, Achieve Results, and Systems Transformation. The LEADS framework provides a formal way to conceptualize the qualities necessary for successful health care leadership. [14] The domains and capabilities overlap and interact with each other, reflecting that leadership is constantly evolving and changing. [14] The LEADS framework has been proposed as a common language to assist individuals in navigating health care leadership and creating change. [15] Specifically, knowledge transfer, evidence sharing, autonomy, shared values, and clarity were qualities of the group's work that were highlighted as being integral to success. The existing collegial and trusting relationships between key stakeholders allowed for interdisciplinary collaboration, rapid mobilization, and identification of issues and finally facilitated the success of the project itself.

An important finding of this case study is that there was no formal leadership position given to any individual in the project. The work was done by leaders who worked in parallel, engaging with each other as needed while each working within their own organization and area of expertise to achieve a common goal of the project. This provided some difficulty to uniformly apply the LEADS Framework which is designed for more formal team environments.

Participants did not report that there were clear guidelines set for conflict resolution as suggested by LEADS. [26] The framework does not specify the content of conflict guidelines but highlights the importance of having an agreed upon process of dealing with conflict before it occurs. In the case where a group involves individuals that are less well known to each other, these guidelines would help to navigate the socio-political environment of the team. Conflict resolution skills and training have been shown to improve both teamwork and productivity as well as job satisfaction. [31] One participant did provide information on the mutual respect between team members that has in the past lead to the successful navigation of disagreements, but that none occurred, to their knowledge, during the project. There were no disagreements mentioned in the data and so the need for these guidelines was low. In a situation of collaborative groups with mutual trust and respect, working more in parallel, it may not be practical or necessary to establish formal guidelines.

Most participants also did not describe ongoing assessment and evaluation following completion of the project, although this is a central feature of the CART-mifepristone implementation research. Although metrics and formal performance assessments may not have been relevant to the project, post-project reflection and documentation in the form of an after-action review may have been helpful. Completion activities are recognized as an important, but often neglected, aspect of project management. [32] They serve to identify potential areas for improvement and to provide a guide to other individuals working in other domains who might benefit from the expertise and experience of the groups involved. [32] This study fills part of that gap. The CART team has ongoing plans to evaluate how provision of medical abortion in Canada has changed as a result of changing regulations. Importantly, most participants

were not aware of these ongoing assessments and so communication from the CART team could be improved.

One participant raised recognition as an important opportunity for improvement. The domain of Systems Transformation includes the importance of supporting innovation, which includes recognition of the work of team members, as a way to encourage ongoing engagement with system change. Participants did identify that the team was very strong in fostering a community where all stakeholders were invested in the outcome and felt that their work mattered. Recognition of leaders, especially in a health care setting, has been acknowledged as a key component to retaining and supporting individuals in leadership roles. [33]

Our analysis provides evidence that the leadership skills of the team members were a key contributor to the removal of mifepristone restrictions. However, we do acknowledge that leadership alone is not the only requirement for health policy change. The process often consists of complex interactions of policy, problems, and politics as described by Kingdon. [34] It is important to note that the success of the project was also likely supported by opportune timing in which there was a problem that needed solving, a favourable political climate, and the appropriate policy makers with motivation to change regulations in BC. [34]

Abortion services can be a potentially contentious or sensitive topic for health care systems. A shared vision and belief in the importance of access to abortion care for Canadians was another driving force behind the success of the project. However, participants' lobbying for change to provisions in prescribing practices also relied on the beliefs that these restrictions were in direct disagreement with current prescribing practices and policies in BC. This overarching belief was likely important to success as it meant the project focused on a flaw in policies regulating prescribing and dispensing medications broadly, not abortion specifically.

## Strengths and limitations

Our interviews were completed in 2019 and the project concluded in 2017. This delay between action and interview likely contributed to participants' limited recall of certain details. However, all participants did provide a very similar sequence of events. Through our analysis of news media and journal articles we were able to corroborate timelines. Additionally, there was no conflict reported between group members or instances where the group had to alter its strategy or approach. It is possible that the time delay described above made it difficult for participants to remember any negative aspects of working together that may have occurred. As well, given that the overall goal of the project was achieved, the participants may have been less likely to remember aspects that did not work well from a team/leadership perspective despite our specific probing questions in the interviews. Our inclusion of analyses of correspondence between participants and minutes from meetings, neither of which reflected negative or unsuccessful stages in the project, mitigates against this recall bias. There have been many examples where, despite strong leadership, policy change has not been successful. Recently, in Quebec, Canada, efforts to improve access to medical abortion have been unsuccessful. [35]

## Conclusion

The results of analysis of the interviews and documents provide valuable insight into the workings of a successful group committed to a common goal. Document analysis also provided a clear timeline of the work completed and suggested that strong communication between team members was key to success. The existing collegial and trusting relationships between key stakeholders allowed for interdisciplinary collaboration, rapid mobilization, and identification of issues as well as facilitating a successful change to mifepristone dispensing regulations in Canada. A lack of awareness of the CART plans for post-project assessment, such as this study,

was identified as a possible limitation to the work of the team. In future iterations of this team, we could incorporate planned assessments/evaluations and thus strengthen dynamics, especially when a desired outcome is not achieved.

In addition to improving future work within BC, the analysis provided may be helpful to other national or international organizations by identifying the key leadership attributes that contributed to success. Medical abortion continues to be difficult for individuals to access globally, with many of the same (or more severe) restrictions to provision in place in settings around the world. [36] These qualities could be cultivated and encouraged in local teams and combined with other national health care leadership frameworks in use in non-Canadian jurisdictions, such as Health LEADS Australia or the UK's Healthcare Leadership Model to augment the results to improve applicability. [15] The study also provides guidance on the mobilization of research data to create change through increased public awareness, which was very important to the project and can be used in international settings. Through showing an example of leadership success we hope to provide a blueprint for other teams working to drive change in important areas. We demonstrated that existing collegial and trusting relationships between key stakeholders allowed for interdisciplinary collaboration, rapid mobilization, and identification of issues that facilitated successful Canadian global-first deregulation of mifepristone dispensing.[36]

## Author Contributions

**Conceptualization:** Brigid Dineley, Sarah Munro, Wendy V. Norman.

**Data curation:** Brigid Dineley.

**Formal analysis:** Brigid Dineley, Sarah Munro.

**Investigation:** Sarah Munro.

**Methodology:** Brigid Dineley, Sarah Munro, Wendy V. Norman.

**Resources:** Wendy V. Norman.

**Supervision:** Wendy V. Norman.

**Writing – original draft:** Brigid Dineley.

**Writing – review & editing:** Brigid Dineley, Sarah Munro, Wendy V. Norman.

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
