## [Decision Letter · Decision Letter 0]

7 Oct 2019

PONE-D-19-22742

Leadership for Success: Health Policy Change in Canada

PLOS ONE

Dear Ms Brigid Delaney

Thank you for submitting your manuscript to PLOS ONE. After careful consideration, we feel that it has merit but does not fully meet PLOS ONE’s publication criteria as it currently stands. Therefore, we invite you to submit a revised version of the manuscript that addresses the points raised during the review process.

We would appreciate receiving your revised manuscript by 1 November 2019. To enhance the reproducibility of your results, we recommend that if applicable you deposit your laboratory protocols in protocols.io, where a protocol can be assigned its own identifier (DOI) such that it can be cited independently in the future. For instructions see: http://journals.plos.org/plosone/s/submission-guidelines#loc-laboratory-protocols

We look forward to receiving your revised manuscript.

Kind regards,

Helen Schneider, MBChB, MMed, PhD

Academic Editor

PLOS ONE

Journal Requirements:

1. We note that you have indicated that data from this study are available upon request. PLOS only allows data to be available upon request if there are legal or ethical restrictions on sharing data publicly. For information on unacceptable data access restrictions, please see http://journals.plos.org/plosone/s/data-availability#loc-unacceptable-data-access-restrictions.

Reviewers' comments:

Reviewer's Responses to Questions

**Comments to the Author**

1. Is the manuscript technically sound, and do the data support the conclusions?

Reviewer #1: Yes

Reviewer #2: Yes

2. Has the statistical analysis been performed appropriately and rigorously? 

Reviewer #1: N/A

Reviewer #2: N/A

3. Have the authors made all data underlying the findings in their manuscript fully available?

Reviewer #1: Yes

Reviewer #2: Yes

4. Is the manuscript presented in an intelligible fashion and written in standard English?

Reviewer #1: Yes

Reviewer #2: Yes

5. Review Comments to the Author

Reviewer #1: OVERALL COMMENT: This is a worthwhile and interesting study.

- Would be useful to know what the legislation does say in Canada about abortion.

- To know the power of different bodies in this process. To understand better what value the LEADS framework provides in comparison to other framework for these issues.

ABSTRACT

Line 47: Mention is made of a Canadian First Global deregulation: This is unclear – this not in main body of article and needs justification that this is a global first.

INTRODUCTION

- The introduction could explain better point one above i.e.

Provide background to what the legislation says about abortion choice and rights in Canada. It would be value for the reader to know what the actual law in Canada says about abortion - is it allowed under any circumstances, at all stages of pregnancy? Any restrictions? Any health personnel needing to authorize an abortion at any stage of pregnancy before women can undergo a procedure? Any age restrictions?

- Identifies lack of access to medical abortion as a gap, inequity – should mention some other areas that have barriers. Should be worded so that this is not the only solution to lack of access to abortion.

- Description of Canadian setting could be expanded on a bit.

- Are there any other countries that have dealt with this kind of problem, given that a number of countries have medical abortion?

- If LEADS is an acronym, write out in full first time a it is used – if not then explain where it comes from.

- Value of LEADS framework can be expanded on in comparison with other frameworks

METHODS

Line 140: use of a consistent guide: Can the authors explain the rationale for this as one of the benefits of qualitative research is its flexibility and not necessary using a consist guide.

RESULTS

Lines 249 and 250: Would be good to show to expand a bit more on the data analyzed on these points.

Lines 337 and 338: “ knowing who to use..” An expansion of this explanation is needed for this to b clearer for the reader.

Line 379: PDSA cycle: More description of this model needed. And which components of the model this project adhered to.

Lines 386 and 387: More information on this needed on the interactive cycles of engagement with reference to the data is needed to understand how this worked and had an important effect.

Lines 400 and 401:” leaders are aware….” Need to expand to explain to what end – was this for championing and orchestrating change as in the sub-title?

Lines 443 and 444: more information on the conflict resolutions suggestions in LEADS is needed for the reader to understand this aspect better.

Line 467: recognition as an important factor for improvement: More explanation is need to clarify what this means and where is this in the findings.

DISCUSSION

- Although focus in on the system and leadership it would be valuable in the discussion or conclusion to note limitation of the study - the role of providers in implementing etc. Other barriers to abortion access either not

LIMITATIONS

Mention any limitations of this study and not only emerging from areas not dealt with in the interviews or the data..

CONCLUSION

- Could link this better to use in the international context.

TYPOS

Check for missing punctuation and need for spacing between words in some of the text.

Reviewer #2: This is an interesting and important case study. The paper is well written and structured overall.

The title of the article is quite vague – I would recommend reflecting the focus on mifepristone in the title.

Within the introduction discussion about why this analysis was conducted is missing. From a policy perspective why is this important know? i.e. investigate leadership skills, etc.?

Proof reading is needed overall (watch spacing, use of full stops, etc.)

Methods

Why was the LEADS Framework determined to be an appropriate analytical tool to assess policy change? Were policy frameworks such as Kingdon’s policy windows considered?

The framework purports that leadership is key for policy change. In your opinion is this true? Are there other factors which are also important and may have helped? Discussion about the role and importance of leadership in relation to policy change is needed.

It is not typical to describe the individual team member roles within the methods section in such detail and usually methods are described more broadly.

There is a lot of detail in the methods section which may not be necessary. For example, example interview guide questions are usually not included. The discussion at the beginning of the results section is a little repetitive.

Page 9 Line 158-159 – sentence is unclear.

Results and Discussion

The result section is well structured. A diagram including the main thematic areas under the LEAD framework may be helpful.

Overall, the paper takes a very positive spin on the topic and role of leaders in policy change. While I’m not disputing this, as there is little to no discussion of challenges or what didn’t work well it makes the reader think that there may have been some selective bias in representation of results.

I would have liked to have seen the findings situated more within the wider literature in the discussion section.

What are the main limitation of the study?

Are there any recommendation or key mess

ages that can be included within the discussion/ conclusion? Why is the information worthwhile? What can be done with it? What does it mean for broader policy change?

The focus on mifepristone policy change is very interesting. Is there anything that can be said about policy change in relation to social issues which may be viewed as contentious? The fact that the stakeholders were able to implement change in regards to a potentially sensitive topic is impressive and can definitely be learned from. What might take away from the study in their quest to change policy? Are the findings only applicable to Canada or more broadly, including within the US?

6. PLOS authors have the option to publish the peer review history of their article (what does this mean?). If published, this will include your full peer review and any attached files.

Reviewer #1: Yes: Diane Cooper, Professor, School of Public Health, University of the Western Cape, Cape Town, South Africa.

Reviewer #2: No

---

## [Author Response · Author response to Decision Letter 0]

9 Nov 2019

Thank you for your consideration of our study. In this letter, we respond to each point from the reviewers, with the original comments indicated first and our response indicated subsequently.

Reviewer Comments

Reviewer 1 

OVERALL COMMENT: This is a worthwhile and interesting study.

- Would be useful to know what the legislation does say in Canada about abortion.

- To know the power of different bodies in this process. To understand better what value the LEADS framework provides in comparison to other framework for these issues.

a) Thank you for your positive feedback on our study. We appreciate your perspective to know more about the context in Canada and agree this will improve our paper. We hope you will find that these suggestions have been addressed in our response to the comments below. See response c.

ABSTRACT

Line 47: Mention is made of a Canadian First Global deregulation: This is unclear – this not in main body of article and needs justification that this is a global first.

b) A reference has been added. Specifically: Berer M, Reconceptualizing safe abortion and abortion services in the age of abortion pills: A discussion paper, Best Practice & Research Clinical Obstetrics and Gynaecology. This additional citation discusses the global first of the recent deregulations in Canada.

Introduction: 

- The introduction could explain better point one above i.e. Provide background to what the legislation says about abortion choice and rights in Canada. It would be value for the reader to know what the actual law in Canada says about abortion - is it allowed under any circumstances, at all stages of pregnancy? Any restrictions? Any health personnel needing to authorize an abortion at any stage of pregnancy before women can undergo a procedure? Any age restrictions?

- Identifies lack of access to medical abortion as a gap, inequity – should mention some other areas that have barriers. Should be worded so that this is not the only solution to lack of access to abortion.

- Description of Canadian setting could be expanded on a bit.

- Are there any other countries that have dealt with this kind of problem, given that a number of countries have medical abortion?

- If LEADS is an acronym, write out in full first time a it is used – if not then explain where it comes from.

- Value of LEADS framework can be expanded on in comparison with other frameworks: 

c) We have added a more detailed explanation of the legal status of abortion in Canada, specifically, a clearer description of what it means to be decriminalized and the absence of regulations around gestational age and medical board approval. (lines 52-67 on page 4 ). As well, we have expanded on the inequitable access to abortion in Canada, pointing out three main issues with access (cost, knowledge and geography) and clarify that medical abortion provides a potential solution to the geographic location of abortion providers. (lines 69-85 on page 4) We have clarified how the US and Australia in particular have faced similar regulatory barriers to medical abortion. (lines 91-95 on page 5) The reason for choosing the LEADS framework has been provided and the acronym has been explained. (lines 123-152 on pages 6-7) 

Methods: Line 140: use of a consistent guide: Can the authors explain the rationale for this as one of the benefits of qualitative research is its flexibility and not necessary using a consistent guide.

d) We have clarified that we chose to use a “semi structured” interview guide in order to address the same theoretical domains from the LEADS framework with each participant. The interview guide was designed to address the LEADS domains and so we wanted to ensure that each person had an opportunity to speak to each domain. However, if a particular domain was not relevant to a participant (as indicated during the interview) we asked probing follow up questions to better understand their initial response. In this sense, our qualitative approach was flexible and adaptable. Please see lines 191-192 on page 9.

Results:

Lines 249 and 250: Would be good to show to expand a bit more on the data analyzed on these points

e) We have added that The CART team was able to use knowledge translation to take their research data that indicated that the restrictions in place were limiting provision of medical abortion and bring it to a new set of knowledge users. Please see lines 333 to 335 on page 14.

Lines 337 and 338: “knowing who to use.” An expansion of this explanation is needed for this to be clearer for the reader. 

f) We have added that this participant was referring to the ability to recognize team members’ strengths and areas of expertise and make the best use of them, instead of confining everyone only to their formal roles within an organization as being creative and necessary to success. Please see lines 425 to 427 on page 19.

Line 379: PDSA cycle: More description of this model needed. And which components of the model this project adhered to. 

g) We have added that the PDSA cycle is framework created by the National Health System (NHS) in the UK to guide implementation of novel changes in a regulated and controlled fashion. The system is important in leadership because it allows for innovation to be both supported and introduced in a way that encourages ongoing evaluation. Please see lines 470 to 473 on page 21.

Lines 386 and 387: More information on this needed on the interactive cycles of engagement with reference to the data is needed to understand how this worked and had an important effect. 

h) We have clarified that the iterative cycles of engagement consisted of research data being generated and then shared with stakeholders and other knowledge users throughout the country (e.g. media). Please see lines 479 to 483 on page 21.

Lines 400 and 401: “leaders are aware…” Need to expand to explain to what end – was this for championing and orchestrating change as in the sub-title? 

i) The relationship has been described in more detail explaining that leaders are aware and understanding of the relationships between stakeholders and other knowledge users and individuals within the health system. A leader can support and stimulate engagement from a variety of different groups in order to generate change. Please see lines 499 to 501 on page 22.

Lines 443 and 444: more information on the conflict resolutions suggestions in LEADS is needed for the reader to understand this aspect better.

j) We have clarified that the LEADS framework does not specify strategies for conflict resolution, but does suggest that there is a pre-defined process for resolving conflict when it arises. Please see lines 545 to 547 on page 24.

Line 467: recognition as an important factor for improvement: More explanation is need to clarify what this means and where is this in the findings. 

k) We have added that the domain of Systems Transformation includes the importance of supporting innovation, which includes recognition of the work of team members, as a way to encourage ongoing engagement with system change. Please see lines 570 to 573 on page 25.

Discussion/Limitations: - Although focus in on the system and leadership it would be valuable in the discussion or conclusion to note limitation of the study - the role of providers in implementing etc. Other barriers to abortion access either not

LIMITATIONS: Mention any limitations of this study and not only emerging from areas not dealt with in the interviews or the data.

l) A limitations section has been added. The important point that medical abortion does not address all of Canada’s issues with respect to access to abortion was clarified in the introduction section and so has not been reiterated in this section. Please see lines 596-609 on page 26

Conclusion: Could link this better to use in the international context.

m) We have added information on why the study is important nationally and international and what role the results could have for other teams. Please see lines 627-638 on page 28.

Typos: Check for missing punctuation and need for spacing between words in some of the text.

n) We have carefully proofread the manuscript and made corrections throughout.

Reviewer 2 

This is an interesting and important case study. The paper is well written and structured overall.

The title of the article is quite vague – I would recommend reflecting the focus on mifepristone in the title.

a) Thank you for your positive feedback on our study. We welcome the suggestion to improve our title. The title has been changed to: Leadership for Success in Transforming Medical Abortion Policy in Canada

Within the introduction discussion about why this analysis was conducted is missing. From a policy perspective why is this important know? i.e. investigate leadership skills, etc.?

b) We have added an explanation of why the project was undertaken and its value to both provide guidance for other national and international teams as well as for improving work within the CART team in the future. Please see lines 116 to 120 on page 6.

Proof reading is needed overall (watch spacing, use of full stops, etc.)

c) We have carefully proofread the manuscript and made corrections throughout.

Methods: 

Why was the LEADS Framework determined to be an appropriate analytical tool to assess policy change? Were policy frameworks such as Kingdon’s policy windows considered?

d) We have clarified why we chose to use the LEADS framework over alternative policy frameworks including that we wished to specifically explore the role leadership played in the work that was done to remove the provision restrictions and we wished to use a framework specifically designed for the Canadian health care system. Please see lines 139 to 152 on page 7.

The framework purports that leadership is key for policy change. In your opinion is this true? Are there other factors which are also important and may have helped? Discussion about the role and importance of leadership in relation to policy change is needed.

e) We do feel that leadership is necessary to create an environment where evidence based change and be introduced and sustained. We acknowledge that there was likely a “window of opportunity” as described by Kingdon that contributed to this project’s success and have added this to the discussion. Please see lines 142 to 148 on page 7 and lines 579-585 on page 26.

It is not typical to describe the individual team member roles within the methods section in such detail and usually methods are described more broadly. There is a lot of detail in the methods section which may not be necessary. For example, example interview guide questions are usually not included. The discussion at the beginning of the results section is a little repetitive.

f) We have made the methods section more succinct and removed repetition from the beginning of the results section. The Journal Article Reporting Standards for Qualitative research recommends the following details be included in a qualitative research paper:

“The relationships and interactions between researchers and participants relevant to the research process and any impact on the research process (e.g., was there a relationship prior to research, are there any ethical considerations relevant to prior relationships). Describe coders or analysts and their training, if not already described in the researcher description section (e.g., coder selection, collaboration groups). Describe how the researchers’ perspectives were managed in both the data collection and analysis (e.g., to limit their effect on the data collection, to structure the analysis). Describe questions asked in data collection.” (See: APA Journal Article Reporting Standards. Information Recommended for Inclusion in Manuscripts That Report Primary Qualitative Research. Available at: https://apastyle.apa.org/jars/qualitative.)

For this reason, we have left in the descriptions of relationships between the investigators as well as the examples of interview questions. We also did not remove the description of bias mitigation as we felt this was important for transparency. Finally, we also left in the description of CART’s work as it is referenced throughout the paper.

Page 9 Line 158-159 – sentence is unclear.

g) This has been clarified, each interview was assessed through the lens of each domain of the LEADS framework. Please see lines 216-217 on page 10.

Results and Discussion

The result section is well structured. A diagram including the main thematic areas under the LEAD framework may be helpful.

h) This has been added. Please see Figure 3.

Overall, the paper takes a very positive spin on the topic and role of leaders in policy change. While I’m not disputing this, as there is little to no discussion of challenges or what didn’t work well it makes the reader think that there may have been some selective bias in representation of results.

i) This is an important point. We have added a limitations section where we address that there was minimal report on challenges. We also have discussed situations in Canada with medical abortion where attempts to improve access have been unsuccessful. Please see lines 596-609 on pages 26-27.

I would have liked to have seen the findings situated more within the wider literature in the discussion section.

j) Thank you for this excellent suggestion. We have discussed failed policy attempts in the new limitations section (lines 596-609 on pages 26-27), we have also discussed how the findings could be modified based on other leadership frameworks in use in other countries. (lines 626-637 on page 28). A literature search was unable to identify any studies that have taken a similar approach to a policy change analysis. 

What are the main limitation of the study

k) A limitations section has been added. Please see lines 596-609 on pages 26-27

Are there any recommendation or key messages that can be included within the discussion/ conclusion? Why is the information worthwhile? What can be done with it? What does it mean for broader policy change? The focus on mifepristone policy change is very interesting. Is there anything that can be said about policy change in relation to social issues which may be viewed as contentious? The fact that the stakeholders were able to implement change in regards to a potentially sen¬¬¬sitive topic is impressive and can definitely be learned from. What might take away from the study in their quest to change policy? Are the findings only applicable to Canada or more broadly, including within the US? 

l) Thank you again for your reflections and this indication of an area for us to improve our paper. The conclusion/discussion has been expanded to take into consideration these ideas. Please see lines 587-593 on page 26 and lines 611 to 637 on pages 27-28.

---

## [Decision Letter · Decision Letter 1]

16 Dec 2019

Leadership for Success in Transforming Medical Abortion Policy in Canada

PONE-D-19-22742R1

Dear Dr Brigid Dineley,

We are pleased to inform you that your manuscript has been judged scientifically suitable for publication and will be formally accepted for publication once it complies with all outstanding technical requirements.

With kind regards,

Helen Schneider, MBChB, MMed, PhD

Academic Editor

PLOS ONE

Additional Editor Comments (optional):

Reviewers' comments:

Reviewer's Responses to Questions

**Comments to the Author**

1. If the authors have adequately addressed your comments raised in a previous round of review and you feel that this manuscript is now acceptable for publication, you may indicate that here to bypass the “Comments to the Author” section, enter your conflict of interest statement in the “Confidential to Editor” section, and submit your "Accept" recommendation.

Reviewer #1: All comments have been addressed

2. Is the manuscript technically sound, and do the data support the conclusions?

Reviewer #1: Yes

3. Has the statistical analysis been performed appropriately and rigorously? 

Reviewer #1: N/A

4. Have the authors made all data underlying the findings in their manuscript fully available?

Reviewer #1: Yes

5. Is the manuscript presented in an intelligible fashion and written in standard English?

Reviewer #1: Yes

6. Review Comments to the Author

Reviewer #1: (No Response)

7. PLOS authors have the option to publish the peer review history of their article (what does this mean?). If published, this will include your full peer review and any attached files.

Reviewer #1: Yes: Diane Cooper PhD, School of Public Health, University of the Western Cape, Cape Town, South Africa

---

## [Editor Report · Acceptance letter]

23 Dec 2019

PONE-D-19-22742R1 

Leadership for Success in Transforming Medical Abortion Policy in Canada 

Dear Dr. Dineley:

I am pleased to inform you that your manuscript has been deemed suitable for publication in PLOS ONE. Congratulations! Your manuscript is now with our production department. 

With kind regards,

on behalf of

Dr. Helen Schneider 

Academic Editor

PLOS ONE